

# Contributions to a neurophysiology of meaning: the interpretation of written messages could be an automatic stimulus-reaction mechanism before becoming conscious processing of information

Roberto Maffei, Livia S. Convertini, Sabrina Quatraro, Stefania Ressa and Annalisa Velasco

A.L.B.E.R.T. Research group, ARPA-Firenze—Cultural association, Firenze, Italy

## ABSTRACT

**Background.** Even though the interpretation of natural language messages is generally conceived as the result of a conscious processing of the message content, the influence of unconscious factors is also well known. What is still insufficiently known is the way such factors work. We have tackled interpretation assuming it is a process, whose basic features are the same for the whole humankind, and employing a naturalistic approach (careful observation of phenomena in conditions the closest to "natural" ones, and precise description before and independently of data statistical analysis).

**Methodology.** Our field research involved a random sample of 102 adults. We presented them with a complete real world-like case of written communication using unabridged message texts. We collected data (participants' written reports on their interpretations) in controlled conditions through a specially designed questionnaire (closed and opened answers); then, we treated it through qualitative and quantitative methods.

**Principal Findings.** We gathered some evidence that, in written message interpretation, between reading and the attribution of conscious meaning, an intermediate step could exist (we named it "disassembling") which looks like an automatic reaction to the text words/expressions. Thus, the process of interpretation would be a discontinuous sequence of three steps having different natures: the initial "decoding" step (i.e., reading, which requires technical abilities), disassembling (the automatic reaction, an unconscious passage) and the final conscious attribution of meaning. If this is true, words and expressions would firstly function like physical stimuli, before being taken into account as symbols. Such hypothesis, once confirmed, could help explaining some links between the cultural (human communication) and the biological (stimulus-reaction mechanisms as the basis for meanings) dimension of humankind.

Corresponding author
Roberto Maffei,
roberto@robertomaffei.it

## INTRODUCTION

Human–environment interactions entail conscious knowledge, i.e., the attribution of meanings (semantic aspect) to the incoming signals and stimuli. Interpretation, namely the operation through which the meaning is attributed, is still widely unknown. A specific difficulty is represented by natural language, although it has been studied almost since the dawn of humankind, with researches ranging from the ancient rhetoric (for example, *Geymonat, 1970*; *Barthes, 2000*; *Perelman, 1981*) to the most recent approaches complementing linguistics with biology and neuroscience (for example, *Zuberbühler, 2005*; *Locke, 2009*; *Stekelenburg & Vroomen, 2012*). Messages are (or, at least, they appear) made up just of words; however, understanding a message always goes far beyond its words.[1] The available data does not give definite answers to the researchers' questions; our field research intends to bring some contributions to such endeavour.

### Research lines and ideas: a synthetic overview

The available scientific literature is so wide to prevent, inside the boundaries of our work, an exhaustive analysis. However, a rapid survey is sufficient to reveal some trends, the first of which is the accelerating extension of these studies towards the field of science. Another trend, thanks to the extraordinary development of technology and informatics, is the enhancement of the studies that explore interpretation inside the brain and its neural processes.

All this considered, we can roughly outline a picture with two main scientific research lines, the first of which can be named **Mind-centred approaches** and can be synthesized as follows. Understanding/interpretation is based on abstract (conceptual) knowledge. Incoming information is provided through the body (perception) but the "mind"[2] processes inputs at symbolic level, turning them into propositional representations in the brain and understanding them in terms of concepts. The answer to the inputs (reaction) is based on such comprehension and is shaped as a command to some effectors (typically the motor system). Knowledge is the result of a sort of computation; the mind is separated from the body and rules it. The role of the motor system is totally passive.

The second research line can be named **Body-centred approaches** and can be synthesized as follows. Understanding/interpretation is attained through a motor reaction of the body that instantiates understanding or, at least, co-exists with conceptual knowledge. When an external stimulus/signal is perceived, it is firstly "understood" through a motor reaction which is automatic, involuntary and based on "mental maps" that are motorial, not (or not only) propositional. Understanding is a sort of motor experience that goes along with conscious (rational) information processing; the body is not detachable from the mind and can drive it. The role of the motor system is active and decisive for understanding.

The main features of the first group theories are synthesized in some recent works, for example, *Zipoli Caiani, 2013* (Chapters 1 and 2); *Ferrari & Rizzolatti, 2014* (specially Pag. 2); *Gallese, 2014* (specially Pag. 2, with the concept of ontological reductionism); *Pulvermüller et al., 2014* (specially Introduction and Fig. 1).[3] The range of these theories goes

[1] Material regarding the attempts to explain human communication and the questions of meaning and interpretation is really countless. Specific works will be indicated within the manuscript. Taking linguistics apart, we make reference to *Pettigiani & Sica (2003)* for a review (in Italian) of psychological main approaches; *Krauss & Fussell (1996)* for a wide survey from the perspective of social psychology.

[2] We will not enter the disputed question of mind, its existence, its nature and its relationships with the body in general and the brain in particular. For a first level of delving further into this subject: on the one hand, the early survey of *Sperry (1952)*; on the other hand, the more recent works of *Marcus (2004)*, *Rose (2005)* and *Zeki (2010)*. In the context of our Introduction, the "mind" is simply intended as a factor which, by following some theoretical positions, totally controls body through functions that differ from biological processes.

[3] The last three works (*Ferrari & Rizzolatti, 2014*; *Gallese, 2014*; *Pulvermüller et al., 2014*) are ascribable to the theories of the second group; nonetheless, they are cited also here because contain particularly clear syntheses of the opposite field positions. Ahead in the text we will describe a mirror-case (*Hickok, 2009*).

from the merely mechanical (and naïve) hypotheses of psychoneural isomorphism (*Sperry, 1952*, pp. 293–294), and those inspired by the first electronic computers (*Newell, Shaw & Simon, 1958*), to the various I.P. (information processing) models (*Massaro & Cowan, 1993*) and current cognitive science positions (*Negri et al., 2007*; *Mahon & Caramazza, 2008*; *Mahon & Caramazza, 2009*). The shared concept is that information is essentially processed in a linear and unidirectional sequence, based upon a functional (besides the anatomical) separation among sensory, associative and motor areas of the brain cortex (for a general presentation and discussion, see also *Rizzolatti & Sinigaglia, 2006*, Chapter 1, specially pages 20–22; for a synthesis of the cognitivist paradigm, see *Gallese, 2000*, page 27).

The second group of theories (the body-centred ones) can be traced back, at least, to XIXth Century, up to the works of *Lotze (1852)* (cited in *Rizzolatti & Sinigaglia, 2006*) and *James (1890)*, which present reflections on the relationships between perception and action. Other philosophers came after,[4] up until a new series of neurophysiological studies appeared in the second part of XXth Century.[5] Such researches gathered evidence that the sequential processing theory and the supposed motor system passive role are untenable. A leap ahead has probably been accomplished with the discovery of mirror neurons (*Di Pellegrino et al., 1992*) and the related following studies (for example, *Gallese, 2000*; *Rizzolatti & Craighero, 2004*; *Iacoboni et al., 2005*; *Rizzolatti & Sinigaglia, 2006*). According to this theory, understanding would be firstly attained through a motor reaction of the body, "immediately and automatically".[6] Cognition would be "embodied".

Embodiment of cognition, and its consequences on knowledge and interpretation process, are the object of a lively scientific debate well exemplified in *Hickok (2009)* (direct reference to *Rizzolatti, Fogassi & Gallese, 2001*). Imagine someone pouring a liquid from a bottle into a glass: by following the embodied cognition hypothesis, an observer can "embodily" understand such action since, thanks to his mirror neurons, he undergoes a motor reaction "as if" himself was actually pouring (by the way, such reaction does not turn into any actual movement, it remains virtual). However, that pouring "could be understood as *pouring, filling, emptying, tipping, rotating, inverting, spilling* (if the liquid missed its mark) or *defying/ignoring/rebelling* (if the pourer was instructed not to pour)..." (see *Hickok, 2009*, page 1240, italics by the author). Such examples, in our opinion, well represents the crucial point: the scientifically evident automatic reaction that instantiates embodied cognition does not explain the whole process of interpretation, and the attribution of a conceptual meaning seems to have a different nature. Thus, we have either scientific evidence of embodied cognition or daily-life experience of scattered conceptual interpretations; can these two visions be conciliated or are they alternative? And which one can actually account for the field observations?

In the few last years, the hypotheses based on the mirror neurons discovery have been refined, for example through the concepts of Mirroring mechanisms (MM) and Embodied simulation (ES) (*Gallese, 2005*; *Gallese, 2006*; *Gallese, 2007*; *Gallese, 2008*; *Gallese, 2009a*; *Gallese et al., 2009*; *Gallese & Sinigaglia, 2011a*; *Ferri et al., 2011*; *Marino et al., 2011*; *Gallese & Sinigaglia, 2012*; *Ferrari & Rizzolatti, 2014*; *Gallese, 2014*). About the ongoing dispute, a summary and a state-of-the-art outline can be found in *Zipoli Caiani (2013)* and one of the

---

[4] Some special mentions about the philosophers: (*Mach, 1897*), in particular pages 1–8 (on the relationship between scientific knowledge and perceptual experience of physic world), pages 15–17 (a famous example on subjectivity of perspective) and pages 93–95 (sense organs as active elements of perception, fine-tuned through experience, rather than as passive receptors); (*Poincaré, 1902*), especially Chapter 4 (on the relations between geometrical space and "representative," i.e., perceptual, space); (*Poincaré, 1908*), Part I, specially pages 52–63 (phenomenology of a mathematical discovery and the role of sensitivity and aesthetic feeling); (*Merleau-Ponty, 1965*), particularly Part II (with special regards to introduction chapter, on the impossibility to have a knowledge of the environment that is independent of the body experience).

[5] Some special mentions: (*Sperry, 1952*), especially pages 299–300 on the relationships among perceptions and ideas; (*Jeannerod et al., 1995*; *Liberman & Whalen, 2000*; *Fowler, Galantucci & Saltzman, 2003*).

[6] We are intentionally employing the words "immediately and automatically": they are typically used in describing the mirror-system working.
most interesting documents is a forum (*Gallese et al., 2011*) inside which the most delicate and controversial questions are widely debated.

## Experimental research involving language

Theoretically, the divergence between cognitivist and embodied cognition approaches can be synthesized as follows (for further reference see, for example, *Bedny et al., 2008*; *Rizzolatti & Fabbri-Destro, 2008*; *Goldman & De Vignemont, 2009*; *Gallese, 2011*; *Gallese & Sinigaglia, 2011b*; *Bedny et al., 2012*): cognitivism upholds the sequential processing idea, i.e., cognition would be the result of perception (the sound of a spoken message as well as the sight of written words) followed by the symbolic processing of what perceived (turning the spoken or written words into their meanings) followed by a reaction (typically, but not exclusively, a motor one). Oppositely, the embodiment theories uphold the concept of direct connections among cortical sensorial and motor areas ("sensorimotor grounding" of cognition, *Guan et al., 2013*). Namely, the perceived spoken or written words would trigger a motor reaction and would be mentally represented also in a motor, rather than a purely conceptual, way. In this sense, cognition would be embodied.[7]

From a technical slant, the two research lines tend to privilege different laboratory approaches: cognitivist field frequently engages the noun-verbs dissociation problem, studying it through researches on cortically damaged, selectively impaired patients (for example, *Crepaldi et al., 2006*; *Arévalo et al., 2007*; *Moseley & Pulvermüller, 2014*; *Gallese, 2014*). Conversely, the embodied cognition theorists mainly go searching for the connections between language and its motor correlates, one well-known of which is the ACE (Action-sentence Compatibility Effect; for example, *Vitevitch et al., 2013*; *Horchak et al., 2014*). Such studies are frequently carried out through neuroimaging works (for example, *Tettamanti et al., 2005*; *Aziz-Zadeh et al., 2006*; *Speer et al., 2008*; *Aziz-Zadeh & Damasio, 2008*).

It is interesting to note that, beyond their important differences, both cognitivism and embodiment research use, in laboratory experiments, words and short phrases isolated from every contexts (see, for example, *Bedny et al., 2008*; *Bedny et al., 2012*, especially the Method sections; for some critical reflections about the question, *Pulvermüller et al., 2014*, specifically Pag. 80, Chapter 7). Such approach entails that meaning is an intrinsic feature of words, something embedded inside them, and that interpretation consists in extracting it (actually, the verb "to extract" is overtly used in scientific publications, for instance *Mahon & Caramazza, 2011*).

## About some recent trends

In the end, it is worth mentioning a recent specialised research field inside psychophysics, in which researchers investigate cognition and semiosis through probabilistic models (*Chater, Tenenbaum & Yuille, 2006*; *Ingram et al., 2008*; *Tenenbaum et al., 2011*), applying the Bayesian inference to reproduce mental processes and to describe them through algorithms (*Arecchi, 2008*; *Griffiths, Kemp & Tenenbaum, 2008*; *Bobrowski, Meir & Eldar, 2009*; *Arecchi, 2010c*; *Perfors et al., 2011*; *Fox & Stafford, 2012*). Such concepts are currently in use also in the Artificial Intelligence (AI) field[8]; in addition, some studies make reference to deterministic chaos (*Guastello, 2002*; *Arecchi, 2011*) and some others to Gödel's

[7] Such embodment, inside the same embodied cognition field, can be conceived in different ways: it can stand alone, *per se* resolving the problem of knowledge ("sensorimotor processing underlies and constitutes cognition", *Guan et al., 2013*), or can be a "motor representation" that accompanies conscious knowledge processes (the two kinds of knowledge proposed by Gallese, for example in *Gallese et al., 2011*; see also *Gallese, 2014*).

[8] The origins of Artificial Intelligence (AI) studies can be traced back to the Thirties and the works of Alan Turing on a possible "intelligent machine". About the origins, see *Leavitt, 2007*, chapters 6 and 7, and *Turing, 1950* (the original work of Alan Turing). About the "Turing test" (testing the ability of distinguishing humans from computers through exchanging written messages) see a journalist's account in *Christian (2012)*. Some materials about recent research lines, closer to our article's topics (like machine learning and natural language or image interpretation), can be found in *Mitchell (1997)*, *Menchetti et al. (2005)*, *Mitchell (2009)*, *Khosravi & Bina (2010)* and *Verbeke et al. (2012)*.

incompleteness theorem as a limit to the possibility of understanding cognition "from inside" (given that, while studying cognition, we become a system that investigates itself).[9]

## Methodological aspects and our approach

There are two main reasons why the question of interpretation and meaning has not yet been scientifically solved, the first of which is that there are still structural obstacles of technical and ethical nature.[10] The second main reason is the complexity of natural language (its "equivocal" nature, see *De Mauro, 2003*[11]), which is usually overcome through studying interpretation isolated from the interpreting organism and employing simple stimuli (for instance *Bedny & Caramazza, 2011*).

In field experiments, researchers who capitalise on the existence of mirror neurons intentionally favour a naturalistic-like approach, letting the observed macaque monkeys freely interact with available objects, rather than stimulate them with selected artificial stimuli only (*Rizzolatti & Sinigaglia, 2006*, p. 3; in addition, about the reductionism question and the distinction between methodological and ontological reductionism, see *Gallese, 2000*, p. 26, and *Gallese, 2009b*; *Gallese, 2010*). However, their approach has been also criticized (*Pascolo & Budai, 2013*). About the naturalistic-like approach, we had in our background two works about interactions inside online collaborative groups (*Maffei, 2006*; *Maffei, Cavari & Ranieri, 2007*) which let us appreciate the potential of scientific observation in real world-like conditions.

On these bases we set up our approach. We set two objectives for our research: (i) to understand the process of interpretation (i.e., how messages in natural language are turned into meanings by receivers) as it works in real conditions, and design a structural model in order to adequately represent it; (ii) to produce a first check of the formulated hypothesis. We tried a naturalistic approach; this means, first, that a phenomenon must be carefully observed and precisely described in conditions the closest to "natural" ones (natural conditions = the way and the contexts in which the phenomenon usually manifests). Second, it means that observation and description must precede analysis, being carried out independently of it. In such approach, the role of the observers is critical, either if they are involved in or external to the phenomenon. In our research, we have employed 102 observers of the first kind (the sample) and 5 (the authors) of the second one; this way, we have collected 102 self-reports (participants' answers to a specially designed questionnaire) and worked out one analytical report (our research) about interpretation.

## METHOD

This was not a clinical trial and no experiments were conducted on the participants. In addition, no personal data was collected or in any way used in the survey, and verbal informed consent was obtained. The Ethics Committee for Scientific Research of the Association ARPA-Firenze gave its approval either to the research design or to the informed consent procedure. Further details related to method, sampling and ethical aspects can be found in the Supplemental Information (SI, from now on), Sections 0, 1 and 3.

---

[9] See *Goldstein (2006)* for a popular-scientific coverage about Gödel and his theorem; *Leavitt, 2007*, chapters 2 and 3, for a particularly clear synthesis of the theorem and its genesis (in connection with the *Entscheidungsproblem*, i.e., the "decision problem").

[10] About the technical difficulties of data collecting: experimental techniques used on macaque monkeys (electrode direct insertions inside single neurons) return very accurate measurements, but on small brain cortex surfaces. About the ethic difficulties: those techniques are almost impossible to be used on humans, and only indirect techniques as fMRI (functional Magnetic Resonance Imaging), MEG (Magnetoencephalography), PET (Positron Emission Tomography) or TMS (Transcranial Magnetic Stimulation) are systematically employed. They cover wider brain cortex surfaces but with inferior accuracy; moreover, they present difficulties with regards to instrument positioning and image interpreting. For a survey of these difficulties see (*Rizzolatti & Sinigaglia, 2006*), chapters 2, 6, 7, and (*Rizzolatti & Vozza, 2008*), *passim*. A recent line of research is investigating the connections among single neurons activity and the total effects detectable through indirect techniques (see *Iacoboni, 2008*, chapter 7). In addition to all this, data interpretation and comparing are intrinsically difficult, given the differences in macaque and human brain cortex and the associated problem of identifying reliable correspondences.

[11] *De Mauro (2003)* states that natural language is "equivocal" in etymological sense: from Latin *aeque vocare* (to name (different things) in the same way). That means: a same word can be used to refer to different meanings and different words can be used to indicate the same meaning.

**Table 1 Main features of the sample (total sample).** The table provides a quantitative description of the total sample with regards to age (left columns), education level (central columns) and employment (right columns) of the participants; see Legends for the used symbols. Data is shown either as values or in percentage and split down by gender (M, males; F, Females).

| | Age | | | | | | Education | | | | | | Employment | | | | |
| | M | | F | | | | M | | F | | | | M | | F | | |
| Bin | Val. | % | Val. | % | Tot | Bin | Val. | % | Val. | % | Tot | Bin | Val. | % | Val. | % | Tot |
|---|---|---|---|---|---|---|---|---|---|---|---|---|---|---|---|---|---|
| A | 10 | 23.8 | 32 | 76.2 | 42 | El | 1 | 25.0 | 3 | 75.0 | 4 | A | 16 | 47.1 | 18 | 52.9 | 34 |
| B | 11 | 36.7 | 19 | 63.3 | 30 | Dg | 18 | 46.2 | 21 | 53.8 | 39 | B | 6 | 85.7 | 1 | 14.3 | 7 |
| C | 7 | 46.7 | 8 | 53.3 | 15 | Gr | 18 | 30.5 | 41 | 69.5 | 59 | C | 6 | 31.6 | 13 | 68.4 | 19 |
| D | 9 | 60.0 | 6 | 40.0 | 15 | – | – | – | – | – | – | D | 1 | 20.0 | 4 | 80.0 | 5 |
| – | – | – | – | – | – | – | – | – | – | – | – | E | 5 | 17.2 | 24 | 82.8 | 29 |
| – | – | – | – | – | – | – | – | – | – | – | – | F | 3 | 37.5 | 5 | 62.5 | 8 |
| Tot | 37 | | 65 | | 102 | Tot | 37 | | 65 | | 102 | Tot | 37 | | 65 | | 102 |

**Notes.**

Legend (*age*): A, 18–29 yy; B, 30–39 yy; C, 40–49 yy; D, 50 yy and over.
Legend (*education*): El, Elementary level; Dg, High school degree; Gr, Graduates/post-graduates.
Legend (*employment*): A, Line workers; B, Managers; C, Graduated technicians/professionals; D, Artisans/Entrepreneurs; E, Students; F, Unemployed/others.

## Materials and procedure/1: the sample

Our research plan has been based on two main assumptions: first, interpretation is a process, rather than a single operation; second, the process has the same basic (structural) universal characteristics. The rationale of our sampling was based on such assumptions: according to our objectives, we focused on the reconstruction and understanding of the process, rather than on sample features. Thus, the sample representativeness (for example, with respect to Italian people), as well as its social feature balance, were less critical; from an extreme point of view, it could be sufficient that the sample members would belong to human species. Operatively, we gathered our random sample through selecting only Italian language native speakers, all adult; we strived to reach a reasonable balance about gender and student/worker conditions. Further details (the procedure we used to randomize the sample included) can be found in SI, Section 6; the results are presented in Tables 1–3.

The total sample (Table 1) results slightly imbalanced with regards to gender (women exceed men), education (Graduates/Post-graduates exceed High-school degree granted members) and employment (students/unemployed exceed employed members). For these reasons, even though social features balance is less relevant in our work, we have selected more homogeneous sub-samples from the total sample, in order to verify our analyses every time it turned out necessary. The first sub-sample ("AGE," Table 2) is exclusively composed by people over 29 years-old (60 members); the second one ("EMPLOYMENT", Table 3) is exclusively composed by employed people (65 members).

## Materials and procedure/2: the case

The main operative instruments through which we have implemented our naturalistic-like approach (further details in SI, Section 0) are the case and the questionnaire. We challenged our randomly selected sample of 102 adults with a real world-like written communication case, using complete and unabridged message texts and collecting

**Table 2 Main features of the sample (sub-sample "Age," >29yy).** The table provides a quantitative description of the sub-sample "Age" (only participants 30 years, and over, old) with regards to age (left columns), education level (central columns) and employment (right columns) of the participants; see Legends for the used symbols. Data is shown either as values or in percentage and split down by gender (M, males; F, Females).

| | Age | | | | | | Education | | | | | | Employment | | | | |
| | M | | F | | | | M | | F | | | | M | | F | | |
| Bin | Val. | % | Val. | % | Tot | Bin | Val. | % | Val. | % | Tot | Bin | Val. | % | Val. | % | Tot |
|---|---|---|---|---|---|---|---|---|---|---|---|---|---|---|---|---|---|
| A | / | / | / | / | / | El | 1 | 25.0 | 3 | 75.0 | 4 | A | 14 | 46.7 | 16 | 53.3 | 30 |
| B | 11 | 36.7 | 19 | 63.3 | 30 | Dg | 12 | 52.2 | 11 | 47.8 | 23 | B | 6 | 85.7 | 1 | 14.3 | 7 |
| C | 7 | 46.7 | 8 | 53.3 | 15 | Gr | 14 | 42.4 | 19 | 57.6 | 33 | C | 6 | 37.5 | 10 | 62.5 | 16 |
| D | 9 | 60.0 | 6 | 40.0 | 15 | – | – | – | – | – | – | D | 1 | 25.0 | 3 | 75.0 | 4 |
| – | – | – | – | – | – | – | – | – | – | – | – | E | 0 | 0.0 | 2 | 100 | 2 |
| – | – | – | – | – | – | – | – | – | – | – | – | F | 0 | 0.0 | 1 | 100 | 1 |
| Tot | 27 | | 33 | | 60 | Tot | 27 | | 33 | | 60 | Tot | 27 | | 33 | | 60 |

Notes.
Legend (*age*): A, 18–29 yy; B, 30–39 yy; C, 40–49 yy; D, 50 yy and over.
Legend (*education*): El, Elementary level; Dg, High school degree; Gr, Graduates/post-graduates.
Legend (*employment*): A, Line workers; B, Managers; C, Graduated technicians/professionals; D, Artisans/Entrepreneurs; E, Students; F, Unemployed/others.

**Table 3 Main features of the sample (sub-sample "Employment", job owners).** The table provides a quantitative description of the sub-sample "Employment" (participants with a regular employment only) with regards to age (left columns), education level (central columns) and employment (right columns) of the participants; see Legends for the used symbols. Data is shown either as values or in percentage and split down by gender (M, males. F, Females).

| | Age | | | | | | Education | | | | | | Employment | | | | |
| | M | | F | | | | M | | F | | | | M | | F | | |
| Bin | Val. | % | Val. | % | Tot | Bin | Val. | % | Val. | % | Tot | Bin | Val. | % | Val. | % | Tot |
|---|---|---|---|---|---|---|---|---|---|---|---|---|---|---|---|---|---|
| A | 2 | 25.0 | 6 | 75.0 | 8 | El | 1 | 25.0 | 3 | 75.0 | 4 | A | 16 | 47.1 | 18 | 52.9 | 34 |
| B | 11 | 40.7 | 16 | 59.3 | 27 | Dg | 13 | 52.0 | 12 | 48.0 | 25 | B | 6 | 85.7 | 1 | 14.3 | 7 |
| C | 7 | 46.7 | 8 | 53.3 | 15 | Gr | 15 | 41.7 | 21 | 58.3 | 36 | C | 6 | 31.6 | 13 | 68.4 | 19 |
| D | 9 | 60.0 | 6 | 40.0 | 15 | – | – | – | – | – | – | D | 1 | 20.0 | 4 | 80.0 | 5 |
| – | – | – | – | – | – | – | – | – | – | – | – | E | / | / | / | / | / |
| – | – | – | – | – | – | – | – | – | – | – | – | F | / | / | / | / | / |
| Tot | 29 | | 36 | | 65 | Tot | 29 | | 36 | | 65 | Tot | 29 | | 36 | | 65 |

Notes.
Legend (*age*): A, 18–29 yy; B, 30–39 yy; C, 40–49 yy; D, 50 yy and over.
Legend (*education*): El, Elementary level; Dg, High school degree; Gr, Graduates/post-graduates.
Legend (*employment*): A, Line workers; B, Managers; C, Graduated technicians/professionals; D, Artisans/Entrepreneurs; E, Students; F, Unemployed/others.

participants' interpretations. The case we submitted to the sample (it is fully detailed and documented in SI, Sections 2, 4 and 5) is a fictional piece very close to some real cases the authors had professionally dealt with (the messages are drawn from actual messages and the outlined relationship between the characters has been actually observed). Exactly, this case is an online (via e-mail) interaction between two colleagues (no previous relations between them) having different roles and ranks in the same organization; the two characters are a female employee (XX) and a male professional (the "architect" YY, Project Account for the installation of a heating plant in XX's office). Their interaction

consists (from its start to its end) in exchanging 5 e-mails, 3 of which (Messages #1, #3 and #5) are sent by XX, which starts and ends the interaction, and 2 (Messages #2 and #4) by YY. Such exchange (whose subject is the work-in-progress of the heating plant) can be divided into two phases, during the first of which (Messages #1, #2 and #3) a conflict emerges that will be solved through a special version of the fourth message (sent by YY); the solution of the conflict is confirmed by the last (fifth) message, in which XX declares her satisfaction. A synthesis of the first three messages is the following (further details and a full documentation can be found in SI, Section 4).

### Msg #1 (XX to YY)

*A 67 word e-mail to the Project Account about the installation of the heating plant in her office. She requires an inspection, claiming about "flaws" in the present state of works. Flaws are no better detailed. She also declares she is speaking on behalf of some colleagues and uses the expression: "we would be pleased if, at least once, someone of our Corporation could come here and control…"*

### Msg #2 (YY to XX)

*A brief (48 words) answer of the Project Account in which the regularity of the Project progress is declared. The message ends with the phrase: "at the moment, the progress substantially complies with the chronogram."*

### Msg #3 (XX to YY)

*A 136 words reply in which XX declares herself totally unsatisfied. Her message presents two main features: (i) some minor flaws are listed; (ii) she expresses what resembles an actual threat against YY, in the case he would not take measures (she specifically refers to a hypothetical "waste of public money", given that the Project funding involved public resources).*

Now the conflict is on and the second phase starts: YY prepares a reply to XX's Msg #3 (namely, he prepares the first version, the "H" one, of Msg #4). The label "H" has been used because such version is a "hard" reply; a YY's colleague suggests him a softer version (the "S" one) in order to avoid exacerbating the conflict. YY accepts the advice, he sends the Softer Msg #4(S) to XX and the case ends with the conflict resolution (XX's satisfaction declared in Msg #5). Full-text versions of the Hard Message #4(H), the Softer #4(S) and of Msg #5 are displayed in Table 4; see also SI, Section 5 and Tables S1 and S2, for details about the rationale of the two alternative messages.

## Materials and procedure/3: the questionnaire and the survey

The questionnaire has been the instrument through which we have challenged the sample with the case; it is fully documented in SI, Section 4. The survey has been divided into two phases, following the interaction structure; in the first phase (Questions #1 and #2), we asked participants to interpret the first three messages and to indicate which "concrete elements" of those messages their interpretations had been based on. In the second phase, we submitted them (separately, see SI, Section 3, for details about submission modalities, counterbalancing of "H"-Hard and "S"-Softer message submitting included) the two versions of Msg #4 and asked them (Questions #3 and #4) to give their separate

Maffei et al. (2015), *PeerJ*, DOI 10.7717/peerj.1361

**Table 4  Full-text of message #4 two versions (H/"Hard and S/"Softer") and Message #5.** This table presents the full-text two versions of Message #4, labelled as "Hard" (the original version by YY) and "Softer" (the version suggested by one colleague of his). The full-text final Message #5 is added.

| Message #4/H (the "Hard" version) | Message #4/S (the "Softer" version)—Message #5 |
|---|---|
| From: YY (*Project Account for the heating plant works*) | From: YY (*Project Account for heating plant works*) |
| To: XX (*Employee in one of the offices affected by the works*) | To: XX (*Employee in one of the offices affected by the works*) |
| Cc: ZZ (*Office referent for the works*) | Cc: ZZ (*Office referent for the works*) |
| Sent: … (*date*) (*hour*) | Sent: … (*date*) (*hour*) |
| **Subject**: R: heating plant | **Subject**: R: heating plant |

Message #4/H (the "Hard" version):

Dear Mrs. XX,

I want to premise that, for the sake of a wise management of the work process, intended to optimize the utilization of our Corporation resources (exactly, in order to avoid wasting public money):

   - Before Project start, I asked the Director of your structure (B wing of the building), Dr. KK, to put a specific person in charge of controlling the work's progress;
   - As far as I am concerned, the indicated person is, and will remain, Dr. ZZ;
   - Dr. ZZ carefully planned the project development steps with us;
   - Each office, situated in the B wing of the building, has been already supplied with heating systems (hardware), fully complying with the timetable agreed with Mrs. ZZ;
   - The heating plant is now working, even though in provisional mode.

I do recommend you to send any communication, concerning the mentioned Project, to the specific person in charge of controlling, in order to avoid (as already happened) message exchange with personnel that is not directly and formally involved within the process.

However, I inform you that, at the moment, the works under discussion have been suspended, in order to enable the provisioning of the plant-control software. It will manage automatically the heating system in the offices, including yours, regulating the warm air diffusion (in order, as said above, to reduce any waste of money).

As soon as the software will be installed by the contractor, the works will come to end. By the way, in this phase they should not affect the rooms situated in the B wing of the building at all, but only the thermo station.

All quantitative and qualitative controls, requested by the CHK form (*formal inspection document*), will be carried out after the end of the works and just before their compliance to fixed quality standards will be attested, as prescribed by the current rules.

This said, I have found your objections very interesting. For this reason, once the real existence of the problems you have marked will be assessed, I will certainly solve them as a part of my duty.

Yours sincerely

The Project Account/Arch. YY—(Corporation branch)

Message #4/S (the "Softer" version):

Dear Mrs. XX,

I remember your last message, which I have already answered, and now I really thank you for this new one. In fact, we do believe that the attention of our colleagues, on field operating with structures and plants we provide, is fundamental to complete our tasks at best.

In order to optimize our contribution, I have been since the beginning asking for a unique person in charge of controlling the works, accounted for your office's building. This person is Doctor ZZ (I might have already mentioned her in my previous answer even though, at present time, I am not certain about this). Her duty is to collect all the observations expressed by the staff about the work in progress, then to send it directly to my office. I think you already know her and she is going to receive a copy of the present message. I thought this would make communication easier.

Concerning your request, you can be certain that, so far, our Project has been developed by following all the technical and formal standards prescribed by the current rules. In addition, I inform you that the works are not yet concluded and final checks (along with possible inspections) are about to be carefully planned. Please, inform your colleagues about the existence of a person in charge of control and do not hesitate to contact her in the case of further observations or possible problems. As I said, she will return your indications to us; this way, I assure you they will not be ignored.

Best regards

The Project Account

Arch. YY—(Corporation branch)

**Msg #5**

Thank you very much for your interest and for the information. That was very kind of you and your answer was exhaustive.

Best regards

XX

interpretations. Finally, after submitting Msg #5 (that ends the interaction), we asked them (Final Question) to indicate which of the two versions (the original "Hard" or the colleague suggested "Softer" one), in their opinion, had been actually sent in order to elicit the final answer.

### The data collection rationale

Our peculiar management of the survey and, specifically, of the participants/survey conductors relationship (SI, Section 0, for details) allow us to exclude that participants' answers are intentionally distorted or insincere. Given this, what data did we exactly collect in our survey? In the first phase (Questions #1 and #2) we collected the participants' conscious reports on their interpretations. Naturally, the reports we gathered cannot be considered as reliable descriptions of the "true" interpretation process; rather, they are descriptions of the participants' subjective (conscious) experiences about interpretation. We thought that, even though the link among these conscious accounts and the true process is unknown, the answers could allow us to observe, in a naturalistic-like way, the behaviours associated to the interpretation process. On this basis, we could probably detect enough clues in order to formulate a hypothesis on the deeper "true" process of message interpreting. In other words: we tried an indirect approach given that the interpretation process cannot be directly observed.

In the second phase (Questions #3, #4 and Final Question), we investigated the relationship between the interpretation of a situation and a consequent decision to be made; such decision was the selection, between the original and the colleague suggested versions of Msg #4 ("Hard" and "Softer" versions), of the one capable to solve the case (i.e., to elicit the final Message #5). Our thought was that the consistency between interpretation and the following decision could give us either further clues for a deeper understanding of the interpretation process or elements for checking our hypothesis.

## RESULTS/1: INTERPRETATION AS A MULTI-STEP DISCONTINUOUS PROCESS

The results presented in this Section are based on data regarding the first phase of the XX–YY interaction (Messages #1–3), investigated through the first part of the questionnaire (Questions #1–2). We recall that each question submitted to the sample sent two inputs: at first, participants were requested to freely interpret some aspects of the messages; then, to account for their own interpretations indicating the "concrete elements" on which these were founded. Given that the two kinds of inputs elicit different kinds of data, we will present separate analyses.

### Answers to the first input of the questions: the interpretation scatter

The answers to the first input of the questions show that the interpretations provided by participants are widely scattered. Such scatter can be observed for all messages and for any part of them, even if accurately selected; we have delved further into one of the cases present in our research. Through Question #2, we firstly asked participants if, comparing

**Table 5 An example of interpretation scatter from our research.** Sixty-one individuals (60% of the sample), after having compared XX's Messages #1 and #3, answered "YES" to Question #2 and provided 83 specifications for the changes they had detected in XX's position toward YY. The table classifies the specifications into 4 main categories and provides some examples for each one of them.

| Category | Sub-category | Examples of participants' interpretations |
|---|---|---|
| *Behaviours* (7 answers) | – | XX requests for an intervention She reports flaws She is just sending a duty communication |
| *Emotions* (16 answers) | XX is: | Angry, disturbed, worried, aggressive, discouraged Brave, impatient, afraid |
| *Relations XX–YY* (41 answers) | XX expresses: XX takes a position: XX: | Assertiveness, aggressiveness, superiority, subordination Tough, technical, neutral Demands a solution Recalls YY to his duty Thwarts YY's plans |
| *Message form* (19 answers) | Msg #3 is more: | Concrete, correct, detailed Direct, effective |

Message #3 with Message #1, they found the attitude of XX (the sender) towards YY (the receiver) being changed ('Method' and SI, Section 4 for the message texts; SI, Section 4 for the question full-texts). Then, to the 61 who answered "YES" (60% of the sample), we requested to specify how they would define the new XX's attitude. They provided 83 specifications: 64 stated XX's position as strengthened, 12 as weakened and 7 unchanged (although these seven, too, had answered "YES" to the first part of Question #2). In addition, we can find completely opposing statements in these specifications and we can see that scattering covers very different aspects of the XX–YY interaction (behaviours, emotions and so on, Table 5).

The observed scatter of interpretations can be represented through a "megaphone-shape" picture (Fig. 1): receivers take into account the same information but their final interpretations diverge. Such phenomenon is well known, there is plenty of literature about it.[12] The question is that, even though these observations are common and undisputed, the reasons why this happens remain to be explained.

## Answers to the second input of the questions: the importance of the not-semantic components

We approached these answers by carefully and sequentially reading them (more than once), and distributing them into homogeneous categories. Such an operation was performed by one of the authors, then discussed and shared with the others; its result consisted in the macro-categories presented in Table 6. We observed that many of them seemed independent of the message content and of its semantic aspects; in particular, the "Other elements" category contains items totally unrelated to the text semantics and content (a tight selection is presented in Table 7). One of the most interesting indications

[12] About interpretation scatter, we have quoted an example (taken from *Hickok, 2009*) in our Introduction. In addition, some descriptions, referred to special cases and entailing divergence of interpretations, can be found in *Bara & Tirassa, 1999* (pp. 4–6, communicative meanings as joined constructions); *Sclavi, 2003* (pp. 93–98, the "cumulex" play); *Campos, 2007* (analysis of a historical communication case).

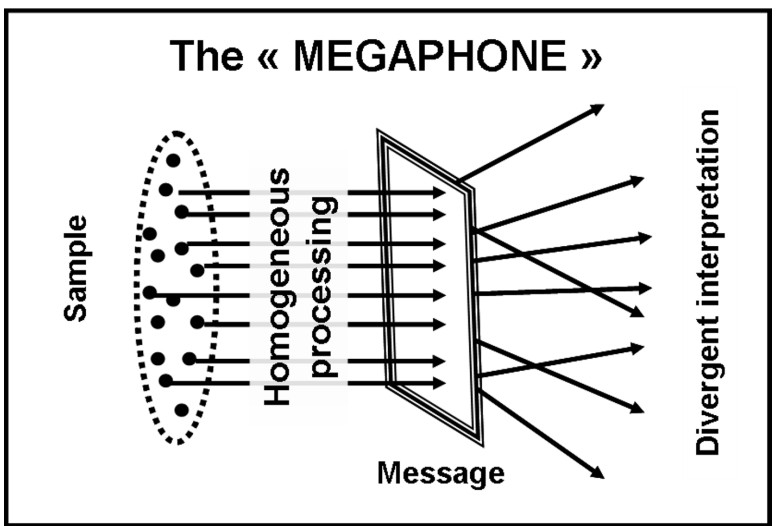

**Figure 1 The "megaphone-shape" model.** If the interpretation of a message should be linked only to the conscious processing of its information content, then we would expect a uniform interpretation, given that the source information is absolutely identical for all the participants. On the contrary, a wide scatter is always observed and its process can be represented with a "megaphone-shape" metaphor: information would be homogeneously processed but differently interpreted.

is the *lack of content* as a "concrete element" (Table 7, final row): how can an information content express a meaning through its absence?

In order to delve further into such matter, we named "components" the categories/sub-categories of the indicated concrete elements and we tried a quantitative analysis. Given that our focus remained on the process, rather than on the sample features, our goal was to provide a rough estimate. Such an estimate was important mainly in relative terms: in case of relative small non-content (non-information) component amounts, we would have to abandon this part of our research. But those amounts were not small. Our analysis of the 1,319 detected components is displayed in Table 8; the indications that clearly focus on the information content constitute only a small minority (around 12%, see Table 8, "%" row, "Cont." column) while references to different text components reach, on the whole, about 65% (Table 8, "%" row, sum of the first five column values). The indications referred to some overall effects of the message represent about 15% of the total. About the meaningless components (void of content *per se*, mere "form" components), their relative amount can be estimated in at least 35% (holding together symbols, incidental passages, other components and grammatical notations).

The proportion of the information content components on the total is very low; even if we sum their relative amount (12.1%) to the indications of full sentences or periods (20.9%, a possible alternative way for referring to the information content) we reach just 1/3 of the total (33%). The question was important and we carried out a further check: we carefully re-examined the filled questionnaires with reference to the information content component. We found out (Table 9) that one half of the sample (51 people) expresses, among the others, at least 1 reference to such component (no similar hint

**Table 6  Macro-categories of the "concrete elements" respondents have indicated as the basis of their interpretations.** The analysis of the answers to the second input of the submitted questions (respondents requested to indicate the "concrete elements" on which the interpretations they provided were based) returned the results displayed in this table. Following the accounts of the participants, their interpretations were based on aspects largely independent of the message information content.

| Category | Sub-categories | Description |
|---|---|---|
| Information content | Information content | Summaries of the message texts and syntheses of their information content, presented through respondent's own words. |
| Meaningful elements | * Words<br>* Phrases/periods | Quotations between double quotes, referred to selected words, full phrases (or parts of them) or periods. Such kind of indications have been provided also through pointing the beginning and the ending word of the quoted strings ("from … to …"). The string length could cover up to a whole paragraph of the message (from a keyboard "Enter" to the following). |
| Incidental passages | Incidental passages | Incidental strings, meaningless *per se*. Such strings were extracted from original full phrases and quoted isolated from the rest. |
| Accessory elements | * Symbols<br>* Titles/salutes<br>* Grammar notations | Complement/accessory parts of the text: punctuation marks[*], personal or professional titles used in the opening, the salutes used in the closing etc.. <br>[*] *In one of the two pilot-sessions of the survey, one message contained an exclamation mark; it was specifically identified, and noted as a meaningful component per se, by one of the participants. For this reason, it was removed in order to limit influencing respondents. In fact, other respondents successively picked up, from questionnaires now bereft of that exclamation mark, quotation marks (used in certain passages of the submitted messages) as a meaningful component per se.* |
| Other elements | Other elements | Items unrelated to the text semantics or to the message content; a tight selection is presented in Table 7. The list is indefinite, given that each item generally appears at low frequency while the range of possible items is extremely widespread. Items of this kind are actually unpredictable; even the *lack of some content* can be focused and reported as a source of meaning (Table 7, final row). |
| Whole message | Whole message | References to some overall effects produced by the message on the participant (see SI Section 8.a, final part, for details). In fact, in this kind of answers, participants state they cannot indicate any "concrete element"; the meaning they have attributed derives from a "general impression" received from the message, from the message's "general tone." |

recordable by the other half). However, only 7 respondents provide a balanced or prevalent amount of indications (50%, or more, of the individual total) about information content. Among them, only one reaches 100%. Such further observation confirms that references to semantic aspects and information content are a definite minority in participants' indications. We added an ultimate control through checking some statistical distributions related to the components, searching for possible imbalances that could contradict our findings. Nothing emerged (for details see SI, Section 10 and Figs. S4–S7).

Following our observations, it seemed that every aspect of even a written message (and even immaterial like an e-mail), regardless of its nature and its intrinsic semantic value, could be treated as a meaningful element of the message, with an extreme degree of scatter among the participants. This was especially surprising because we had used written messages only, bereft of added signals like non-verbal language and context stimuli that usually affect verbal communication (see, for example, *Horchak et al., 2014*, specially the concept of "situated cognition," and *Gibson, Bergen & Piantadosi, 2013*).

**Table 7 A selection of "other elements" that readers may focus on inside the messages.** The table displays a tight selection of the "other elements" (see Table 6, fifth row) focused on by respondents inside the messages. These elements are independent of the information content and, in most cases, of the message text. They are extremely various, indeed unpredictable, and return the impression that the receivers' preferences could be totally rule less.

| Elements | Examples |
|---|---|
| The POSITION of a statement | *XX explains her absence **at the beginning** of Msg #3 to forestall possible criticism.* |
| | *YY scoffs at XX, expressing a little courtesy **just at the end** of Msg #4/H.* |
| The LENGTH of a text | *Msg #4/H **being long** /Msg #5 **being short** have an underlying meaning.* |
| Dotted lists | ***The use of it** in Msg #4/H has a meaning.* |
| Type of lexicon | *The use of **technical words/expressions** implies precision, but also suggests the intention to keep one's distance.* |
| | ***Thanking and reassuring expressions** have détente effects.* |
| The relational or social roles of characters | *Some interpreted Msg #4/H (the "Hard" version) as an attack to XX **being a woman**.* |
| The professional roles of characters | *XX not being an Account, **she would not cheat**.* |
| Grammatical observations | *The **verbs tense is noted** as having an underlying meaning.* |
| LACK of content | *YY **does NOT wonder** why XX requests a control.* |
| | *YY announces a solution **NOT clarifying** what it will be.* |

**Table 8 Statistics on indicated components.** The table displays a descriptive statistical analysis of what the respondents focus on inside the messages. The information content is expressly focused by 12.1% of respondents only ("Cont." column, "%" row). Even if we suppose that reference to complete phrases/periods could actually mean reference to their content, the sum of "Cont." and "Phras." column % totals would amount just to 33% of respondents, again a clear minority.

| Quest. | Sym. | Titl. | Words | Incid. | Phras. | Whole | Cont. | Other | Gram. | TOT | % |
|---|---|---|---|---|---|---|---|---|---|---|---|
| 1-a | 1 | 7 | 46 | 55 | 53 | 16 | 29 | 14 | 4 | 225 | 17.1% |
| 1-b | 1 | 7 | 26 | 53 | 27 | 18 | 20 | 12 | 3 | 167 | 12.7% |
| 1-c | 0 | 6 | 22 | 58 | 34 | 13 | 11 | 12 | 2 | 158 | 12.0% |
| 2 | 4 | 5 | 22 | 52 | 32 | 17 | 34 | 7 | 2 | 175 | 13.3% |
| 3-4/H | 0 | 1 | 13 | 49 | 54 | 35 | 31 | 24 | 2 | 209 | 15.9% |
| 3-4/S | 0 | 22 | 14 | 52 | 48 | 45 | 29 | 5 | 1 | 216 | 16.4% |
| Final | 2 | 14 | 17 | 30 | 28 | 50 | 6 | 22 | 0 | 169 | 12.8% |
| TOT | 8 | 62 | 160 | 349 | 276 | 194 | 160 | 96 | 14 | *1,319* | 100% |
| % | 0.6% | 4.7% | 12.1% | 26.4% | 20.9% | 14.7% | 12.1% | 7.3% | 1.1% | 100% | |

**Notes.**

**Sym.,** Symbols (punctuation marks); **Titl.,** Titles – Salutes (starting/closing expressions); **Words,** Meaningful isolated words/expressions; **Incid.,** Incidental passages, meaningless *per se*; **Phras.,** Complete phrases/periods; **Whole,** General tone of the message; **Cont.,** Information content of the message; **Other,** Other components of the message; **Gram.,** Grammar notations, like verbs tense and similar; **TOT,** Totals; **%,** Percentage on totals.

At this point, we named "disassembling" the observed selective focusing and took two measures. Firstly, we hypothesized a new image for the interpretation process, inverted with respect to the "megaphone-shape" (Fig. 1) one. Our argument was that, if scatter manifests itself in the beginning (scatter of focus), a "funnel-shape" picture (Fig. 2) could be more suitable: people that select the same component are expected to interpret it in very similar ways. Secondly, we picked up from our data an example of disassembling and decided to carry out an in-depth analysis of it.
**Table 9 Sample distribution with regards to components referred to information content.** While answering to the second input of the questions (requesting to indicate the "concrete elements" on which the interpretation was based), just the exact half of the sample indicated, at least once, information content components. In this table, the sample is distributed in bins defined through the percentage that the components referred to information content represent on the personal total of provided indications. Just for 7 people out of 102 the indications pointing at information content balance the others or prevail (50% or more); just 1 person among them indicates information content components only.

| Bins (% on personal total) | N. of respondents | % |
|---|---|---|
| 0% | 51 | 50.0% |
| 1%–24% | 31 | 30.4% |
| 25%–49% | 13 | 12.7% |
| 50%–99% | 6 | 5.9% |
| 100% | 1 | 1.0% |
| TOTAL | 102 | 100.0% |

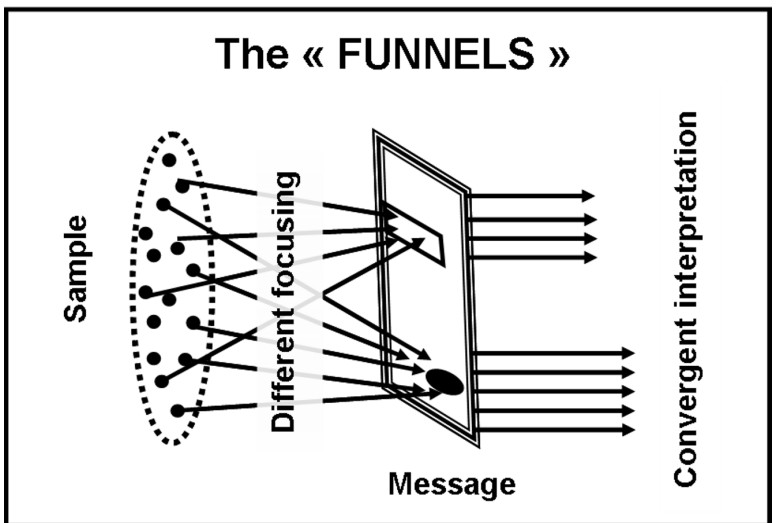

**Figure 2 The "funnel-shape" model.** If the systematically observed scattering of message interpretations would be based on the scattering at "disassembling" step, we could expect that focusing on one same component would be followed by a convergent interpretation of it, as shown in this figure through the metaphor of the "funnels." This is the opposite of the "megaphone-shape" metaphor shown in Fig. 1.

## A disassembling example in detail and a three-step model of the process

Question #1 requested evaluations related to sender-receiver positions and to the relationship between them, on the basis of Messages #1 and #2 (see 'Method' and SI, Section 4, for the message texts). We found out that 53 people (52% of the sample) had quoted an expression the sender (the employee "XX", see 'Method' and SI, Section 2, 4) used in Message #1[13]: she premised her request of a technician inspection with the words "we would be pleased if at least once…" This simple expression, apparently trivial, short (8 words in a 67 word message) and in no way highlighted in comparison with the

[13] The 53 people have reported their interpretations answering Question #1-a (23), #1-b (15) or both the questions (15).

**Table 10 Interpretation scatter referred to one component (the incidental passage of Message #1).** The table displays the result of classifying the interpretations given by a subset of 53 individuals (52% of the sample) to one component of Message #1. These respondents, even though focusing on that same component (the incidental passage "…we would be pleased if at least once …"), have nonetheless dispersed their interpretations. This means that not even the "funnel-shape" model (Fig. 2) could result acceptable.

| Category | Examples of participants' interpretations |
|---|---|
| **"…we'd be pleased…"** (32 quotations) | Aggressiveness; office duty expression; informality; irony |
| | Just a request; sarcasm; highlighting XX's subordinate role |
| | Expression of alternative visions |
| **"…if at least once…"** (17 quotations) | Conflict; doubt on YY's reliability; expression of courtesy |
| | Taunting; request for attention; request for information |
| | A reminder; stimulus to organization top management |
| **"…we'd be pleased… …if at least once…"** (19 quotations) | Expression of XX's fear, because she doesn't feel safe |
| | Insignificant (just a normal office communication) |
| | Complaint/claim |
| | Reprimand/reproach, by XX to YY |
| | XX's clarification request |
| | Information exchange |

rest of the text, has collected 68 quotations (15 people expressed two, see Footnote 13). Then, respondents have interpreted such specific passage in at least 22 different ways, summarized in Table 10.

It seems that focusing on the same component does NOT entail convergent interpretations, that there are TWO levels of scatter instead of one; this could have some important consequences. In terms of metaphors, the previously proposed "funnels" (Fig. 2) were no more suitable; our observations could be much better represented by "hourglasses" (Fig. 3). In terms of process, our observations indicated that the route from the taking into account of a written message (reading it) to the attribution of a conscious meaning to it, could be a sequence of different steps, rather than a unique, homogeneous Input/Output operation (message IN/meaning OUT with the brain cortex as "black-box"processor) like it is tacitly assumed in several current approaches.

Actually, the two actions of focusing on components and interpreting them seem to have different natures. In order to clear this point, we recall an observation reported in the previous sub-section: on the one hand, respondents explain the conscious meanings they attributed through the outcomes of their individual selective focusing (in their answers, they seem to be literally building-up their meanings on the foundations of the picked-up components). On the other hand, they never explain the reasons why they exactly focused on those components: such focusing manifests "immediately and automatically," priming the attribution of a conscious meaning. In addition, if we would assume that focusing and consciously interpreting have the same nature, our reasoning would fall into an infinite regress.[14] So, we can hypothesize the process of message interpretation like a sequence of different steps: how many steps? We must consider that such process actually starts

[14] If the selective focusing on components represents the conscious basis of the attribution of meaning, which could that focusing conscious basis be? And which could be the conscious basis of the conscious basis of that focusing? And so on. A starting point of different nature is anyhow needed.

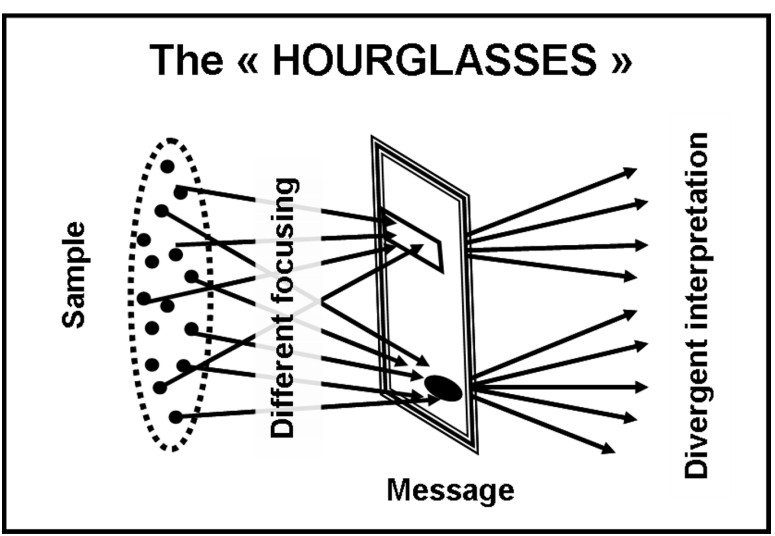

**Figure 3 The "hourglass-shape" model.** This figure displays a metaphor representing the on-field observed process of message interpretation. Two kinds of scatter co-exist, manifesting themselves in sequence: the first one regards dispersion during the focusing on the components ("disassembling"); the second one regards the interpretation of the focused components (conscious information processing).

[15] In our opinion, the process should be the same even in case of oral communication (reading and turning written signs into words should just be replaced by listening to and turning spoken sounds into words).

[16] It is particularly interesting to note that the expression "the fact that…" is spontaneously used by several respondents in their answers. For example, in the collected questionnaires we can find expressions like the following: "the fact that the arguments are presented through a dotted list"; "the fact that XX is referring to public money."

with the reading of the message; this is just a technical step (learned reading abilities in the used language are required) which turns written signs into words.[15] We named it "decoding" and assumed that its outcomes feed the following step (the selective focusing) whose outcomes, in turn, feed the final one (conscious attribution of meaning, based on rational/logical abilities).

In the end, we outlined the model of Fig. 4. The crucial aspect of our hypothesis is the nature of the second step, "disassembling"; on the basis of the presented observations and reflections, we conceive such step as perceptual, not conceptual-logic. The components would act like "physical" stimuli, triggering automatic reactions off ("body" level) in the receivers. We mean: receivers would not consciously recognize the meaning of one component before focusing on it; simply, they would focus on those components suitable to trigger their automatic reactions off.

One last question remains: if a reader reacts to a given component, even though it appears to be meaningless/contentless, we need to identify what, exactly, that reader perceives. We think we can identify it as **the fact that** one of these components is present in the message; it can be considered some meta-information to which readers can automatically react (Table 11). This can clarify the aspect of the incidental passage ("…we would be pleased if at least once…") which triggered the participants' reaction off: *the fact that* XX had (redundantly) placed it at a certain point of her message.[16]

## RESULTS/2: UNCONSCIOUS PROCESSES IN INTERPRETATION–ACTION RELATIONSHIP

The results presented in this Section are based on data regarding the second phase of the XX–YY interaction (Message #4 two versions and Message #5, see Table 4), investigated

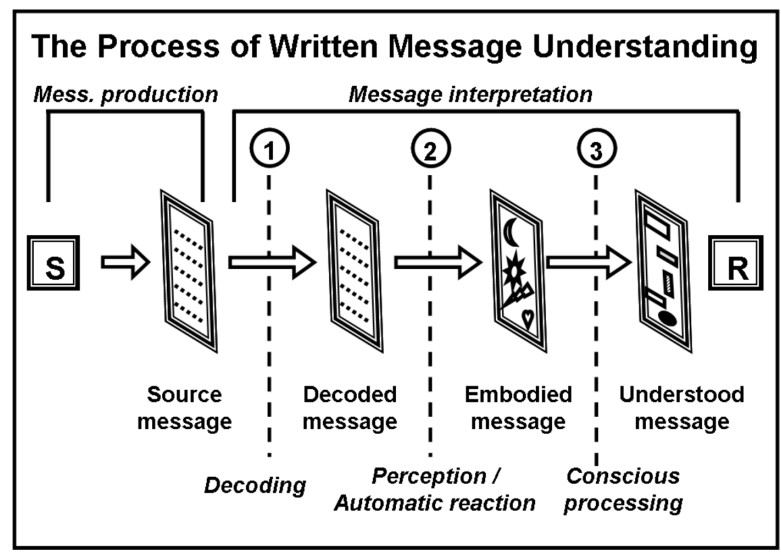

**Figure 4 Scheme of the process of written message interpretation.** S, Sender; R, Receiver; 1-2-3, Progressive steps of the process. This figure presents our hypothesis about how a written message is understood by the receiver. Message production (performed by the sender) is not detailed. The process of interpretation is made up by three sub-processes, in a cascade. The automatic reaction on perceptual basis (step #2) is followed by the conscious information processing (step #3). The step #1 is decoding, given that the words must be, at first, recognized in order to be interpreted.

**Table 11 Examples of possible meta-information stimulus-factors.** The table displays examples, drawn from the filled questionnaires, of a special stimulus-factor inside the messages. The capability of these factors to work as stimuli is not linked to the information they might contain, but to "the fact that" they are present within the message, in a certain form and/or at a certain point (in such sense they represent meta-information to which readers can automatically react).

| Factors | Examples |
|---------|----------|
| Form of address | *Using or not titles indicates formality level* |
| Use of idiomatic expressions | *Sign of familiarity, informality* |
| Regards/greetings form | *Length and presence/absence of thanks are taken into account and interpreted as sign of attention, carelessness, respect, defiance…* |
| Reply quickness | *Courtesy/promptness sign* |
| Use of technical terms | *Sign of intention to keep a distant role* |
| Amount/level of details provided | *Sign of major/minor accuracy or interest* |
| Quantifying information | *Sign of quibbling, coldness* |
| Referring to rules/laws | *Taken as sign of escalation in formality* |

through the questions of the questionnaire second part (Questions #3–4 and Final question). We have submitted to participants two alternative versions of a possible reply to Message #3: the "Hard" original Message #4 and the "Softer" colleague suggested version (in short: Msg #4/H and #4/S; see Table 4 for the full text messages; SI, Section 5 and Tables S1 and S2 for details about the reasons of the proposed alternative). Then, participants were requested to, firstly (Questions #3 and #4), independently interpret the two versions in terms of their effects on XX; secondly (Final question), to choose the version suitable, in

**Table 12 Statistical distribution of the answers to the Final question (choice between the "Hard"/H or "Softer"/S version of Message #4).** The table displays (for the total sample and the two control sub-samples) the frequencies of the answers to the Final question, i.e., the choice between the "Hard" (H) version of Message #4 and the "Softer" (S) one as the solution of the case. A strong imbalance is shown, as indications of Message #4/S overwhelm the Message #4/H ones in all cases.

| Variable | Total Sample | | Sub-sample AGE | | Sub-sample EMPLOYMENT | |
|---|---|---|---|---|---|---|
| | Answers | % | Answers | % | Answers | % |
| *"H" choice* | 26 | *25.7%* | 16 | *27.6%* | 19 | *30.2%* |
| *"S" choice* | 75 | *74.3%* | 42 | *72.4%* | 44 | *69.8%* |
| *Total* | **101** | *100%* | **58** | *100%* | **63** | *100%* |

their opinion, to elicit the final XX's answer (Message #5, that seals the positive ending of the case). Our rationale was the following: the participant's choice could come as a result of the text information conscious processing (cognitivism stance) or as an automatic reaction independent of every conscious processing (embodied cognition stance). In the first case (our "Hypothesis 0"), the final choices should be outcomes of the interpretations given to the messages; thus, they should result somehow correlated with them. In the second case, no correlation, or a different kind of correlation, should be found (our "Hypothesis 1"). The problem was how to assess such correlation.

## The coherence between interpretation and choice

Firstly, we displayed (Table 12) the choices indicated by the sample members and found out a strong imbalance between the "Hard" and the "Softer" version of Message #4. Secondly, we compared the interpretations of Message #4/H (the "Hard" one) with those of Message #4/S (the "Softer" one; Table 4 for full-text messages). Source data (opened answers) was purely qualitative. However, answers were easily classifiable into two main categories: predictions for the message inducing a solution of the case (easing or overcoming, anyhow solving the emerging conflict between the interlocutors); predictions for the message inducing a surge, or escalation, in the conflict. We created the dummy variable "Expected effects" and assigned two values to it: "+" in the first condition; "−" in the second one. Finally, we labelled each questionnaire with two new symbols: one referred to the "Hard" Message #4 (H+ or H−) and one to the "Softer" one (S + or S−). Methodologically, the labelling has been carried out by one of the authors and, independently, by two external persons. The inter-rater reliability has been checked through Fleiss' kappa and resulted 0,95 (excellent rate of agreement).

The combination of the two symbols reports the combined predictions each participant expressed about the effects of the two versions on XX: H+/S+ (both the versions solving the conflict), H+/S− (the "Hard" Message #4 easing the conflict while the "Softer" Message #4 escalating it), H−/S+ (the opposite), H−/S− (both escalating). Dichotomously displaying "H" against "S" predictions (SI, Section 11a and Table S5) returns a clear convergence on combined prediction "H−/S+"; statistical tests

(significance level 5%) confirm that some correlations between the interpretations of the "Hard" and the "Softer" version could exist, even though not all cases result significant (Chi-squared test: $p = 0.029$, total sample; $p = 0.166$, sub-sample "AGE"; $p = 0.038$, sub-sample "EMPLOYMENT"; Fischer's Exact test: $p = 0.043$, total sample; $p = 0.219$, sub-sample "AGE"; $p = 0.064$, sub-sample "EMPLOYMENT"). By cross-checking the combined predictions with the final choice (SI, Section 11a and Table S6) we obtained that the most frequent combined prediction (H−/S+) appears to be strongly associated to the "Softer" message choice; indeed, the significance tests show that some further, stronger relations between combined predictions and choice do exist (Chi-squared test: $p = 0.001$, total sample; $p = 0.035$, sub-sample "AGE"; $p = 0.009$, sub-sample "EMPLOYMENT"; Fischer's Exact test: $p = 0.002$, total sample; $p = 0.027$, sub-sample "AGE"; $p = 0.008$, sub-sample "EMPLOYMENT"). Such results led us facing the core-question related to our hypothesis: given the existence of some correlations between choice and combined predictions, which is its direction? We mean: do the interpretations (the predictions) drive the choice (cognitivism stance) or, oppositely, does the choice precede and somehow drive, or overcome, the interpretations (embodied cognition stance)?

To delve further into such subject, we created a "coherence indicator" starting from the following premises: (i) The final Message #5 clearly indicates XX's satisfaction; therefore, the conflict has come to its end. (ii) Now, let us figure a respondent whose answers to Questions #3 and #4 return a combined prediction H+/S− (the "Hard" Message #4 solving the conflict, the "Softer" one escalating it). Then, we expect that this respondent indicates the "Hard" Message #4 in his final choice. Such combination (H+/S− & "Hard" Msg #4 choice) would represent the maximum coherence level. (iii) If another respondent provides the same combined prediction but chooses the "Softer" Message #4 (combination H+/S− & "Softer" Msg #4 choice), this would represent the minimum coherence level. (iv) Given the natural variability always recorded in human samples, we expected to find also intermediate coherence levels, based on the other possible combinations (H+/S+ and H−/S−). These could also be due to the predictable scattering of interpretations about the final Message #5: someone could interpret it as something different from the sign of the conflict ending (what happened in a fistful of cases).

We defined four coherence levels, increasing from L (low) to LM (low-medium), MG (medium-great) and G (great); the scale is fully represented in SI, Section 11a and Table S7. This way, it has been possible to study the sample distribution with respect to coherence levels (Table 13). The histogram for the whole sample (Fig. 5, data from Table 13) shows the expected shape except for the frequency of the low coherence bin, over-represented. Actually, we expected L frequency to be null or very close to null; anyway, it should result the lowest of all. On the contrary, we found L values higher than the LM ones, representing 12.2% of the sample. The two control sub-samples (right columns of Table 13) show fully comparable features.

At this point, we refined our analysis displaying separately distributions of "H" and "S" choosers; for the reliability of comparison, we excluded data referred to the respondents having just primary education levels (only 4 out of 102 in our sample). Data is displayed

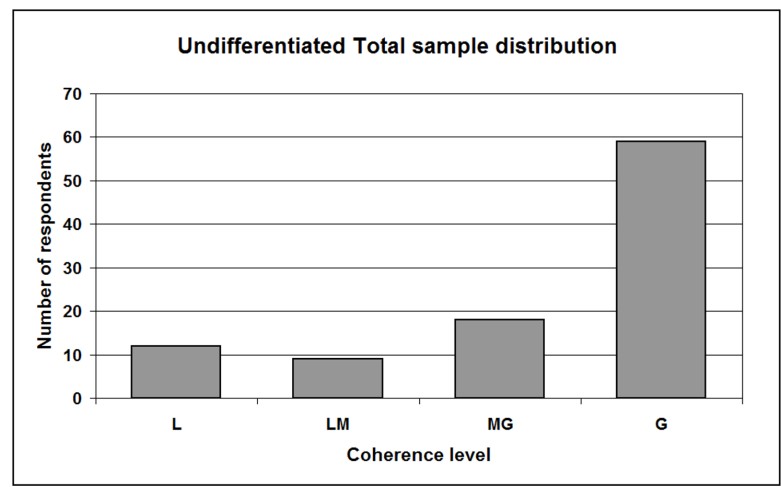

**Figure 5 Sample distribution with respect to coherence levels /undifferentiated total sample.** L, Low; LM, Low-Medium; MG, Medium-Great; G, Great level of coherence. This histogram shows the distribution of ALL respondents according to the coherence (expressed through the coherence indicator) between, on the one hand, their interpretations of Messages #4/H (the "Hard" version) and #4/S (the "Softer" version); on the other hand, their final "H-or-S" choice. Data is shown for the undifferentiated total sample. The L level results over-represented with respect to what expected.

**Table 13 Sample distribution with respect to coherence levels.** The table displays, for the total sample and the two sub-samples "Age" and "Employment," the distribution of participants with respect to coherence levels (see text for concept details; see SI, Section 11a and Table S7 for a display of the scale). The L level results overrepresented with regards to what expected.

| Coherence level | Total sample | | Sub-sample "AGE" | | Sub-sample "Employm." | |
|---|---|---|---|---|---|---|
| | *Values* | *%* | *Values* | *%* | *Values* | *%* |
| **L** | 12 | *12.2* | 8 | *14.3* | 9 | *14.8* |
| **LM** | 9 | *9.2* | 6 | *10.7* | 6 | *9.8* |
| **MG** | 18 | *18.4* | 8 | *14.3* | 9 | *14.8* |
| **G** | 59 | *60.2* | 34 | *60.7* | 37 | *60.7* |
| *Total* | *98* | *100.0* | *56* | *100.0* | *61* | *100.0* |

**Notes.**

L, Low; LM, Low-medium; MG, Medium-great; G, Great level of coherence between predictions and choice; H/S, Versions of Message #4; ±, type of predicted effect (resolution or escalation of the conflict) of the messages on XX.

in Table 14 and complemented in SI, Section 11b, Tables S8 and S9; all the Tables show a surprising asymmetry whose significance is confirmed by Chi-squared test and Fisher's Exact test (in all cases $p < 0.001$). Graphic representations render even better such asymmetry: the total sample histograms (Fig. 6, percent distributions from Table 14) show that the percent frequency of the "Softer" message choosers (white bins) increases regularly from L category to G, reminding (as expected) of certain power, or exponential, curves. Oppositely, the percent frequency of the "Hard" message choosers (grey bins) is arranged in an irregular, almost bimodal shape. We checked these distribution shapes by using many different sub-samples (selection displayed in SI, Section 11b, Figs. S8–S11), included the

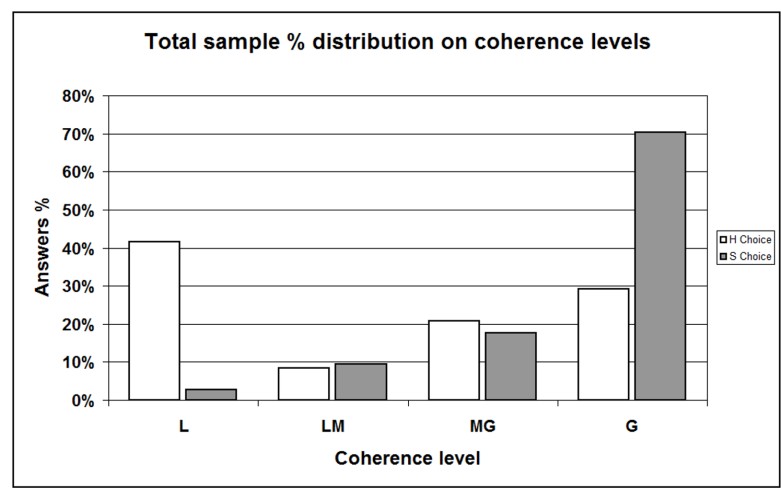

**Figure 6 Sample percent distribution with respect to coherence levels / Comparing "H" and "S" choosers —total sample.** L, Low; LM, Low-Medium; MG, Medium-Great; G, Great level of coherence. This histogram shows the percent distribution of ALL respondents according to the coherence (expressed through the coherence indicator) between, on the one hand, their interpretations of Messages #4/H (the "Hard" version) and #4/S (the "Softer" version); on the other hand, their final "H-or-S" choice. Data is shown separately for "H" and "S" choosers. Distributions result significantly different (Chi-squared test and Fisher's Exact test: $p = 0.000$).

**Table 14 Sample distribution with respect to coherence levels and expressed choice (total sample).** The table displays (for the total sample) the distribution of participants with respect to coherence crossed with the final choice between the "Hard" (H) and the "Softer" (S) version of Message #4. Data shows that the imbalance in the Low coherence bin is ascribable to "H" choosers only. A strong correlation between the two variables "coherence" and "choice" is highlighted: Chi-squared test and Fisher's Exact test return high significance ($p < 0.001$).

| "H" Choosers | | | "S" Choosers | | | Total | |
|---|---|---|---|---|---|---|---|
| *Coherence level* | *Values* | *%* | *Coherence level* | *Values* | *%* | *Values* | *%* |
| **L**(H−/S+) | 10 | *41.7* | **L**(H+/S−) | 2 | *2.7* | 12 | *12.2* |
| **LM**(H−/S−) | 2 | *8.3* | **LM**(H−/S−) | 7 | *9.5* | 9 | *9.2* |
| **MG**(H+/S+) | 5 | *20.8* | **MG**(H+/S+) | 13 | *17.6* | 18 | *18.4* |
| **G**(H+/S−) | 7 | *29.2* | **G**(H−/S+) | 52 | *70.3* | 59 | *60.2* |
| *Total* | 24 | *100.0* | *Total* | 74 | *100.0* | 98 | *100.0* |

Notes.
L, Low; LM, Low-medium; MG, Medium-great; G, Great level of coherence between predictions and choice; H/S, Versions of Message #4; ±, type of predicted effect (resolution or escalation of the conflict) of the messages on XX.

already mentioned "Age" (Fig. 7, data from SI, Section 11b, Table S8) and "Employment" (Fig. 8, data from SI, Section 11b, Table S9) sub-samples. We always obtained the same significant imbalance.

Now, statistical tests and graphic representations clearly indicate the existence of a correlation between the participants' choice and the coherence level; but what about its strength and its direction? In order to investigate the strength, we calculated the odds ratio. Our success item was the L level, our failure items were all the other coherence levels. Using

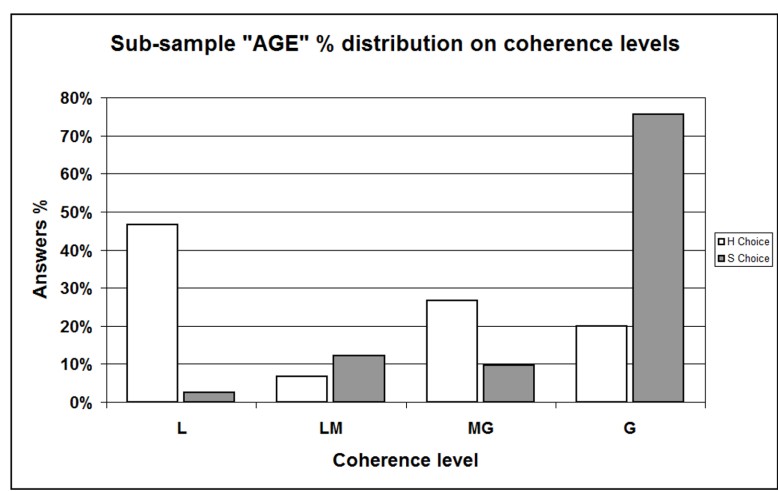

**Figure 7 Sample percent distribution with respect to coherence levels / Comparing "H" and "S" choosers —Sub-sample "AGE."** L, Low; LM, Low-Medium; MG, Medium-Great; G, Great level of coherence. This histogram shows the percent distribution of respondents belonging to sub-sample "AGE" (30 years, and over, old persons) according to the coherence (expressed through the coherence indicator) between, on the one hand, their interpretations of Messages #4/H (the "Hard" version) and #4/S (the "Softer" version); on the other hand, their final "H-or-S" choice. Data is shown separately for "H" and "S" choosers. Distributions result significantly different (Chi-squared test and Fisher's Exact test: $p = 0.000$).

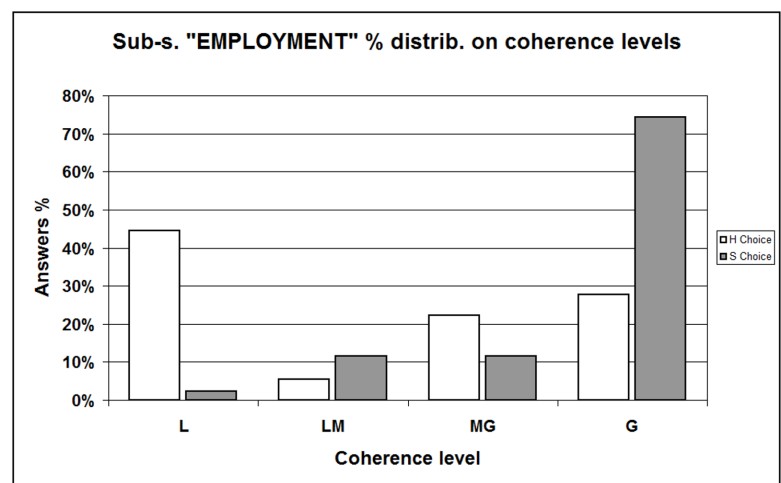

**Figure 8 Sample percent distribution with respect to coherence levels / Comparing "H" and "S" choosers —Sub-sample "EMPLOYMENT."** L, Low; LM, Low-Medium; MG, Medium-Great; G, Great level of coherence. This histogram shows the percent distribution of respondents belonging to sub-sample "EMPLOYMENT" (workers only, students and unemployed excluded) according to the coherence (expressed through the coherence indicator) between, on the one hand, their interpretations of Messages #4/H (the "Hard" version) and #4/S (the "Softer" version); on the other hand, their final "H-or-S" choice. Data is shown separately for "H" and "S" choosers. Distributions result significantly different (Chi-squared test and Fisher's Exact test: $p = 0.000$).

data from Table 14, we can find ODDS1 = 0.417 (the "Hard" version choosers, about 1 success for each failure) and ODDS2 = 0.028 (the "Softer" version choosers, 1 success every about 36 failures). The final result is ODDS RATIO = 25.5 which highlights a strong correlation between the "H" choice and the L coherence level. As much as to say that, if you choose the "Hard" version of message #4, it is much more likely (with respect to the "Softer" version choosers) that your choice is inconsistent with your interpretations of the two messages. About the direction of such correlation (the interpretations precede and drive the choice or the choice is independent of interpretations), we think the first stance is not tenable; indeed, it could be confirmed just in case of general consistency between interpretations and choice.

All this contrasts our "Hypothesis 0": the participants' choice does not seem to come as a result of the text information conscious processing. Then, the choice should be independent of the previous interpretations, what upholds our "Hypothesis 1". After this first conclusion, we set up a second indicator ("block preference" indicator) to further check our hypothesis. For text length reasons, we present details about such indicator, its employment, and relative analysis in Supplemental Information, Section 12 with Tables S10–S13. We found no contradictions with the previous results.

## DISCUSSION

With regards to method, our work showed that studying the interpretation of natural language messages in natural-like conditions can complement laboratory studies based on isolated words/phrases and contribute to a wider comprehension of the phenomenon. With regards to results, the picture outlined through the first part of our work can be synthesized as follows: (i) The interpretation process begins with an operation that looks like a selective and subjective picking up of (or focusing on) the most different components, rather than being a systematic, conscious scanning of the text content. Such behaviour is widely scattered: in the whole research, with regards to each specific message, it is impossible to find two identical combinations of components in participants' answers; (ii) Readers seem to make no distinction among intrinsically meaningful or meaningless components: the meaning they attribute can derive from any "chunk" of the text or from any other text or non-text element arbitrarily chosen; (iii) While the final meaning attributed to the message is justified through the indicated components, no reason (at all, in any cases) is provided for that selection: in the participants' answers, the focused components suddenly appear; they are presented just as "given," and without any doubt.[17] On these bases, we have proposed a three-step model for the interpretation process (Fig. 4); the crucial step is the second one ("disassembling") which, in our hypothesis, is an automatic reaction, out of conscious control. It precedes and feeds forward the conscious attribution of meaning to the message.[18]

If our hypothesis will be confirmed, this means that words are not mere symbols; they are also stimuli (they can act like *physical* stimuli) that trigger automatic reactions off in the receivers.[19] It also means that the third step (conscious attribution of meaning) is fed by the outcomes of the unconscious reaction ("disassembling"), rather than by the original

[17] The unique doubt expressed in the whole research is the following: 1 participant (out of 102) declares uncertainties in his final choice (between the "Hard" version of Msg #4 and the "Softer" one) writing that the final effect could be obtained with both the messages. It must be noted that, with regards to the other questions, this special participant's answers are totally doubt-free.

[18] We have noted that, if disassembling were a conscious passage having the same nature of the following conscious attribution of meaning, the analysis would turn into an infinite regress (see Footnote 14).

[19] Such ambivalence looks interestingly (or just curiously?) similar to what happens in certain physics phenomena like the double nature of light (waves/particles) or the uncertainty about some features of many atomic particles. In those cases, the ambivalence is solved just in the process of measuring the phenomena *Zeilinger, 2012*, for a discussion about the case of photons, and *von Baeyer, 2013* for a recent point of view about such ambivalence); in the case of words, something similar would happen, given that their nature would become evident just in relation with the receiver's reaction.

message; our conscious direct contact with the real world would be prevented, and we would actually attribute conscious meanings just to our automatic reactions to it. In short: through the first part of our work, we have outlined the possible structure of the message interpretation process.

The second part of our work has been designed in a way similar to a social psychology experiment; through it, we have worked downstream with respect to the interpretation process itself, investigating its effects on a consequent behaviour (the final choice); we found out significant imbalances in the coherence between interpretation and choice. Roughly, we can label "rational" the choices that show maximum coherence with the previous interpretations of the two messages (the original "Hard" Message #4, and the suggested "Softer" version); conversely, we can label "irrational" the choices that show minimum coherence. We found that the irrational cases are significantly ascribable to "H" version choosers rather than to "S" version choosers. In other words: the elements provided by interpretations appear insufficient to determine the choice; this means that other factors intervene. Such factors should be unconscious, otherwise they would be declared by at least some participants; in addition, they must have a different and stronger source with regards to the conscious/rational analysis of the message content, otherwise their influence on the choice would not prevail.

The main question is: why, in the decision process, do these factors significantly weigh just in connection with one choice and not with the other one? Further research is needed to find the answer. Provisionally, we think there are two possible hypotheses: (i) The two sub-samples follow different paths in interpreting natural language messages ("Softer" version choosers would base their choices on rational information processing, which would precede action, while "Hard" version choosers would react instinctively and choose before analysing the available information); (ii) The two sub-samples actually follow the same path (automatic reaction preceding conscious information processing, in our opinion) and the difference they show is linked to the differences in their automatic reaction schemes ("Softer" version choosers' reaction would privilege the attention to the relational aspects while "Hard" version choosers' reaction would privilege the content aspects).

## Situating our results in the current research scenario

With respect to the dispute between the stance of cognitivism and the embodied cognition hypotheses, we think our research could be situated in a third position, for two reasons. The first reason is that, while these theories share (even though they come to opposite conclusions) the concept of natural language interpretation as a unique operation, we have seen it as a discontinuous process (three steps of different nature). The second reason is that, in our model, two of the three sub-processes seem to be compatible, separately, with those two theories. We mean: the embodied concept features are akin to our second step ("disassembling"); the cognitivist hypothesis is clearly akin to our third step, (see Fig. 4).

Probably, we can better exemplify this through recovering the example (see *Hickok, 2009*, page 1240) we presented in the Introduction. In our opinion, embodied cognition

hypothesis looks at that described act of pouring in its **purely motorial** nature; conversely, understanding it, for example, as "pouring" or "filling," requires the interpretation of a **situation** which is not limited to the act *per se.* In order to attribute the "pouring" meaning, one must focus on the liquid flow direction (inside to outside the bottle); for the "filling" meaning, one must focus on the glass receiving the liquid; for the "emptying" meaning, one must focus on the amount of liquid inside the bottle. The attribution of conscious meanings should be preceded by the previous, unconscious selection of specific points of view (something closely resembling our "disassembling" step).

Apart from this, if we extend back our literature survey, we can find, for example, that conscious thinking following (rather than preceding) "body" reactions can be traced back up to the hypotheses of the Nineteenth Century philosopher and psychologist William James. In one of his examples (the "James's bear", see *James, 1890*, Chapter XXV), James explains his theory of emotions suggesting that, for example (our synthesis), we do not run away from a bear because we see it, we know it is very dangerous, so we are scared of it and, consequently, we consciously decide to run away (as common sense would sustain). Conversely, we feel we are afraid because (consciously and successively) we discover our body having started a desperate run. In other words: what we call "emotion" is usually intended as a body reaction consequent to the rational processing of consciously perceived environmental stimuli; James suggests that the body reaction immediately follows perception and what we call "emotion" is the consciousness of the new body state (a form of self-consciousness). We are aware that James theory (exactly: James-Lange theory) has been criticized and that alternative theories have been proposed (for example, *Cannon, 1927*; *Schachter & Singer, 1962*); nevertheless, we do refer to it because recent scientific research and reviews seem to suggest some re-consideration of the matter (for example, *Friedman, 2010*).

In the Twentieth Century, we can find the Gregory Bateson's approach to human communication conceived as a system and to the question of the receiver's active role; he uses a strictly formal presentation (see *Bateson, 1976*, in particular Chapter 4.8 on the logical categories of communication, founded on Russel and Whitehead's theory of logical types). In addition, we recall a group of theories and models (some of which expressly refer to Bateson's studies) that tackle the question mainly from a pragmatic slant: the so called "pragmatic models" (*Berne, 1971*; *Watzlawick, Beavin Bavelas & Jackson, 1971*; *Bandler & Grinder, 1981*). Conceived inside a psychoanalytic context, they all put perception and stimuli at the centre of their attention and reverse the relationship between action and thought using action (rather than thought) to induce training and therapeutic effects.[20] We find no important contradictions among our hypotheses and such models; rather, we find complementarity: they show how physical stimuli can act like messages; our results could show that words (even if only written) can act like physical stimuli.

About the relevance of unconscious processes in human behaviour, some fundamental clarification is provided by *Custers & Aarts (2010)* through a review of experimental works; it re-examines the disputed question of the passage from perception to action. The authors compare the traditional positions of Sensory-motor Principle (SMP, for example, *Massaro*

[20] On the one hand, it is worth mentioning a special work coming from NLP founders (*Grinder & Bandler, 1980*): it appears different from the work that founded this theory (*Bandler & Grinder, 1981*) and that has successively been developed by NLP specialists (for example, *Dilts, 2003*). As a matter of fact, that work gives a central role to perception and to physical stimuli (not mediated by language) as a possible communication and therapeutic instrument (see, in particular, the concept of "sensorial anchors" in *Grinder & Bandler (1980)*. On the other hand, we should remind a Watzlawick's work on the modern evolution of psychotherapy (*Watzlawick, 1987*) that represents a severe critic to the classic approach and reverses the relation between action and thought (an Italian translation is retrievable in *Nardone & Watzlawick, 1990*, Chapter 1). In the same *Nardone & Watzlawick (1990)*, see also chapter 2 on perception as one main source of psychopathology.

& Cowan, 1993; for a presentation and discussion about the sequential processing of stimuli conceived as the foundation of human/environment interactions, see also *Rizzolatti & Sinigaglia, 2006*, chapters 1, 2) and Ideomotor Principle (IMP, *Stöcker & Hoffmann, 2004*; *Pezzulo et al., 2006*; *Melcher et al., 2008*; for a synthesis, *Iacoboni, 2008*, Chapter 2, pp. 56–57 of Italian edition). Doing so, they show how certain stimuli (images, solid objects or even written words), intentionally added to an experimental setting, can alter the sample behaviours, even if such stimuli are not consciously detected: "under certain conditions, actions are initiated even though we are unconscious of the goals to attain... (and) goal pursuit can... operate unconsciously" (*Custers & Aarts, 2010*). They also sustain that arguments frequently presented as rational motivations for action are, actually, *ex-post* justifications of unconsciously performed behaviours.

The role of physical stimuli in swaying communication through natural language is confirmed by a series of recent works (for example, *Zhong, Bohns & Gino, 2010*; *Tsay, 2013*; and, for a popular-scientific coverage, *Lobel, 2014*). Further, quite unpredictable factors that can sway message interpretation can be the specific national languages used (for example, *Marian & Kaushanskaya, 2005*; *Costa et al., 2014*) or the metaphors used to express concepts (*Thibodeau & Boroditsky, 2011*; *Thibodeau & Boroditsky, 2013*). Our data is consistent with the outlined scenario in that it confirms the effects of perception-reaction on conscious processing.

## Some possible consequences

Naturally, our results need to be confirmed; once they would be, we can see four main possible consequences. The first one concerns the discontinuous nature of the interpretation process and, specifically, the role of the second step of our model (disassembling) in human communication through natural language: some traditional empirical knowledge would find theoretical bases (for example, in advertising and marketing fields) and a revision of human communication current models would be needed (for example, with regards to mass media and education). Simply, the fact should be taken into account that human communication through natural language could work in a slightly different way than expected and thought up until now.

The second consequence would be the analogical, rather than digital, basis of interpretation. Meaning would be established starting from the body automatic reaction in the "disassembling step," analogically triggered through individual reaction schemes. This could lead to consider natural language expertise as a system of acquired reflexes, what would mean that human beings would "communicate through their body" in a wider and deeper sense than conceived at present (something quite different from mere non-verbal language performances). Such feature could heavily affect the possibility to reproduce human interpretation process on digital computers, regardless of their processing power and data storage capacity. The two systems could result incompatible, rather than simply different. We are not the first who propose such observation (for example, *Arecchi, 2008*; *Arecchi, 2010a*; *Arecchi, 2010b* on the non-algorithmic nature of knowledge and intelligence; *Arecchi, 2010d* on creativity as NON-bayesian process). In

such perspective, if there is any possibility to reproduce the human interpretation process on a computational device, then its model should be the whole human being, not the sole brain cortex. Consequently, what really can prevent present times computers from imitating human thought is not insufficient data processing power or data storage capacity; rather, it is the lack of a special peripheral unit: a human body.

The third consequence could derive from our observations about the taking into account of the message components by the reader, that seems to be performed like a subjective operation, quite arbitrary and unpredictable. If this will be confirmed, the concept of "content of a message" should probably be revised, given that it would result impossible to *ex-ante* define all the contents a reader could detect in a specific message. What is more, as a fourth possible consequence, if mere "form" (aesthetic) components are indifferently taken into account as sources of meaning with respect to the content components, then the difference between form and content fades, leading to a concept of "message" as a unit made up only by *components*, all of them having the same importance (the same *ex-ante* probability of being chosen).

## CONCLUSION

At the end of our arguing about the attribution of meaning, it is worth briefly considering the problem of "what is meaning" (what is the meaning of "meaning"). Beyond the strictly phylosophical, abstract definitions, nowadays we can record attempts to provide operative definitions; for example, *Guastello (2002)*, who considers the sender-receiver couple as a complex system and the meaning like an emergent phenomenon which characterizes it. Our research can lead us to hypothesize another operative definition of "meaning": *The meaning attributed to a message is the receiver's synthetic conscious report (through natural language) on the final state of his/her organism after experiencing the interaction with the message*.

## ACKNOWLEDGEMENTS

We thank Laura Baglietto, Andrea Baldini, Marco Calabrò, Leonardo Cavari, Hasna El-Hachimi, Alessandro Farini, Alessandra Gasperini, Maddalena Morandi, Claudia Santovito, Arabella Tanyel-Kung for their comments.

A special thanks to Fortunato Tito Arecchi for his suggestions; to Andrea Fiaschi and Christina Tsirmpa for their text revision; to Letizia Scrobogna for her contribution to data revision; to Irene Maffei for her final text survey and impact assessment.

### Funding

The authors received no funding for this work.

### Competing Interests

The authors declare there are no competing interests.

## Author Contributions

- Roberto Maffei conceived and designed the experiments, performed the experiments, analyzed the data, contributed reagents/materials/analysis tools, wrote the paper, prepared figures and/or tables, reviewed drafts of the paper, carried out the pilot sessions.

- Livia S. Convertini, Sabrina Quatraro, Stefania Ressa and Annalisa Velasco performed the experiments, analyzed the data, wrote the paper, reviewed drafts of the paper, carried out the pilot sessions; carried out data entry; controlled and harmonized the entered texts.

## Supplemental Information

Supplemental information for this article can be found online at http://dx.doi.org/10.7717/peerj.1361#supplemental-information.

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
