# Peer review of "Contributions to a neurophysiology of meaning: the interpretation of written messages could be an automatic stimulus-reaction mechanism before becoming conscious processing of information"

_PeerJ, doi:10.7717/peerj.1361_

## Round 0.1 · original submission · Major Revisions

· Academic Editor

Major Revisions

Both reviewers note that the study is interesting and potentially adds to the literature on this topic. However, they agree that aspects of the paper are hard to follow, and that the conclusions need to be toned down. In producing a revision of this manuscript, please try to shorten the overall length, provide information about key aspects of your method up front, and take care that all technical terms and abbreviations are defined and easy to follow.

·

Basic reporting

In the abstract several terms (for example reference to the “H” and “S” messages) are discussed without introduction/definition provided to the reader, thus making the text hard to follow (in fact I only discovered what the “H” and “S” messages were when I consulted the supplementary material). I would recommend defining terms early on and/or discussing these kinds of in-depth aspects of the study in greater detail in the methods section. It might also be worthwhile using a term which helps the reader retain what these messages are; for example instead of H use “msg 4 original” and “msg 4 colleague edited” for S.

Generally the manuscript is well written, however there are sections and sentences which are not the same standard as the rest of the manuscript. It is important to stress that these do not detract from the readers understanding of the text, but rather I would like to highlight to the authors that certain sentences and sections do not fully conform to professional standards of courtesy and expression.

This manuscript describes a naturalistic study, and I feel that the format of a standard methods section (participants, design, etc.) would not provide the reader with an accurate description of this study. However this section in the current manuscript would benefit greatly from relevant subheadings as in its current state it is difficult to pick out the key aspects of the studies design and analysis.

Aspects of this manuscript are very dense and hard to understand without consulting the supplementary materials in great depth. I appreciate that this manuscript is already an in-depth and long manuscript. However I feel that it would benefit from a brief discussion of the materials/messages used so that a basic understanding is reached by the reader, so that consulting the supplementary material is optional rather than a requirement.

Within the context of scientific reporting bullet pointed lists should be avoided, which feature a number of times within this manuscript (introduction and discussion). These lists can be incorporated into the text and can be separated without the use of bullet points.

In the discussion too much detail is placed around the origin of the “body” reaction theory. This section forms part of a miniature literature review within the discussion, which should be covered in this degree of detail in the introduction. While I have no issue with bringing in new literature in the discussion, this degree of detail should be reserved for the introduction. This section can be reduced in scope to provide a brief overview (possibly guiding the reader to the introduction) and how these historical accounts directly relate to the study outcomes.

The spelling of Hourglass in Figure 7 is incorrect.

A relatively minor point regards the reporting of the significance values for the results of the chi-squared test; all values should be reported to three decimal places.

Experimental design

The journal requires that “the research must have been conducted in conformity with the prevailing ethical standards in the field”. While it is made clear by the authors that they meet this standard the section describing informed consent and how the participants were recruited could be condensed and/or simplified.

Validity of the findings

I appreciate that the authors recognised the potential differences between their participants, and analysed data by sub-group which demonstrate that the differences observed in the total sample are not driven by the responses of one group. A number of chi-squared tests on all subgroups have not been conducted due to values in that subgroup being less than 5 (or in most cases being equal to zero). I was wondering if the authors had considered using a Fisher’s Exact test to look at potential differences in subgroups with less than expected values. A second related point is whether it is possible to look at differences between these sub-groups (specifically between the larger groups of gender and employment status). You might find that the proportion of favoured interpretations is different between groups, and possibly reveal group differences which might be easier to observe than the individual differences attempted earlier in the paper.

Reviewer 2 ·

Basic reporting

While the topic the manuscript approaches is interesting and the overview of the literature is relevant, the manuscript could benefit from being shortened and written more concisely.


I also strongly recommend that the authors choose a presentation style more typical for scientific articles; e.g., the method section should be short and concise (for example, it should only contain a short statement that the ethics committee approved the study, and that all participants signed informed consent.) Likewise, the abstract should be much shorter (around 200 words).

There are several idiomatic errors throughout the manuscript. I recommend careful proofreading by a native speaker.

p. 9, l. 153: the relevance of the example for describing the theoretical background and the critique of the author don’t become clear to me.

Please include examples of the materials used in the main text.

Experimental design

In the first part of the research, the authors find that messages are interpreted differently by different participants. Why is this finding surprising? And can it really tell us something about human interpretation?
Furthermore, in a quali-quantitative analysis, they find that participants mention many different text features that make them arrive at a certain conclusion. Again, I don’t think it’s surprising that different people focus on different features. Furthermore, an experimental setting is an unnatural setting; we cannot be sure that participants were really motivated to think thoroughly about why they arrive at a certain interpretation, and which features make them do so. And even if they are motivated to do so, it might be very difficult for them to pin down the reasons they arrived at a certain interpretation. Because of that, I think it's difficult to draw a conclusion from the results, especially such strong conclusions as the authors draw at the end of the manuscript.
The second part of the research appears to be a post-hoc analysis of the same data set that had inspired this analysis. Methodologically, it would be better to conduct the same research with a different set of participants and see whether the same findings show for this hypothesis.

Several important aspects of the methodology used aren't explained: Who conducted the quantitative analysis? Were several researchers involved, and are the results reliable across different researchers? The main text should also contain more detailed information about the participants (e.g., age, distribution of gender, whether they were students or not).

Validity of the findings

As stated above, it doesn't become clear whether the results from the quali-quantitative analysis are reliable. The conclusion drawn on the basis of the present data should be much more cautious.

Comments for the author

The aim of the manuscript is to shed some light on the interpretation process. The authors presented texts to 102 participants and asked them to write down their interpretation of the text. The authors found that instead of focusing on the text content, participants tended to concentrate on other features such as physical attributes of the text (text length). The authors conclude from this finding that interpretation is at first a stimulus/response mechanism and that only at a later point, participants switch to interpretation processing. The authors report an experiment in which they seek to examine this hypothesis.
While I find the overall question interesting and worthwhile of investigation, I think that the methods employed can help to give first hints as to how to proceed with further investigation; however, I think they're not enough to answer the questions the authors pose. I recommend the authors take the current findings as a first step to answer their questions, but not (yet) as definite conclusions.

---

## Round 0.2 · Minor Revisions

· Academic Editor

Minor Revisions

Thanks for resubmitting your paper to PeerJ. Reviewer 1 has re-reviewed the manuscript and feels that it is greatly improved. Unfortunately Reviewer 2 declined to re-review.

Since you have a favourable opinion from Reviewer 1, I have provided comments myself and I hope that I will be able to handle the remaining revisions without sending the manuscript back to the reviewers. I must warn you, that I believe that quite a lot of re-writing might still be required for your paper to be accessible to potential readers. In general, I feel that your paper would be easier to understand if the results and background to the research were presented in a much briefer fashion.

In revising this paper, please fully address my comments below and those of Reviewer 1.

I hope you feel you can revise the manuscript along these lines.

With best wishes
Beth Jefferies


Here are my own comments:

The abstract mentions a three-step process but doesn't explain what this is -- please revise.

The Introduction is 8.5 pages: This is too long. Please remove material that is not essential for understanding your argument.

This is a very minor point but the reference to H and S messages isn’t very easy for the reader – writing this out more fully, as Hard and Softer replies, would be easier.

Results from the first part of the research: Please choose a more informative subheading

The results and interpretation/theorising are mixed together which makes it a little hard to work out which of your claims there is good evidence for and which are more speculative.

Line 485: how many discrepant cases. What is inter-rater reliability?

Conclusion section about evolution seems to go a long way beyond the data

There are currently 16 figures and 19 tables! This is a huge number. I don't believe they are all essential. Perhaps non-essential tables and figures could go into supplementary materials or be removed, to make your core argument clearer?

Fig 4: need to explain what the components are.
Fig 9: need to explain what the x-axis categories are.

·

Basic reporting

The reference to the hub and spoke model and the explicit reference to the ATL comes as a bit of a shock in the section describing experimental language research. It’s not really made clear how it relates to this section and the material presented above. One account (i.e. the hub-and-spoke model) proposes that the ATL is heavily involved in stored semantic representation. However this claim is controversial and is a lively topic of debate (for example; Martin, 2007; Martin, Simmons, Beauchamp, & Gotts, 2014). What I would do here is instead of explicitly mentioning the ATL in relation to the hub-and-spoke model I would give examples of different hypotheses and models which attempt to explain “how does the brain code and generate semantic cognition”. For example you could mention the recent work of Pulvermuller (Pulvermüller, 2013) who proposes that there are five separate ‘hub’ regions which represent different aspects of semantic knowledge, related to the motor/sensory features associated with those hubs/regions, alongside the hub-and-spoke model.

At the beginning of the methods section a discussion of the aims and hypothesis is presented. This short section should be at the end of the introduction, and should remove any details to stimuli and participants.

A very small number of sentences in the manuscript still do not read well and detract from the general flow of the manuscript. I have provided suggestions and have attached a version of the manuscript with tracked changes to my review.

Experimental design

No comments.

Validity of the findings

No comments.

Comments for the author

First of all I would like to congratulate the authors in revising this manuscript. This re-submission is more easily approachable to the reader and reviewer, and I feel it provides an improved account of their study. I also appreciate that the authors revised their analysis pipeline and embarked on additional analyses to further strengthen their claims and the outcomes of the paper. However, there are a few minor points which need to be addressed, which I have highlighted in my review.

---

## Round 0.3 · accepted · Accept

· Academic Editor

Accept

Thank you for producing a thorough revision of your manuscript. I found it much easier to follow. Since you have been able to respond to the requested revisions on a point-by-point basis, I can see that you have satisfied all the remaining requirements and I am happy to accept it for publication.